# Is Your Paper Being Reviewed by an LLM? Investigating AI Text Detectability in Peer Review

**Sungduk Yu**      **Man Luo**      **Avinash Madusu**      **Vasudev Lal**      **Phillip Howard**

Intel Labs

{sungduk.yu, man.luo, avinash.madusu, vasudev.lal, phillip.r.howard}@intel.com

## Abstract

Peer review is a critical process for ensuring the integrity of published scientific research. Confidence in this process is predicated on the assumption that experts in the relevant domain give careful consideration to the merits of manuscripts which are submitted for publication. With the recent rapid advancements in the linguistic capabilities of large language models (LLMs), a new potential risk to the peer review process is that negligent reviewers will rely on LLMs to perform the often time consuming process of reviewing a paper. In this study, we investigate the ability of existing AI text detection algorithms to distinguish between peer reviews written by humans and different state-of-the-art LLMs. Our analysis shows that existing approaches fail to identify many GPT-4o written reviews without also producing a high number of false positive classifications. To address this deficiency, we propose a new detection approach which surpasses existing methods in the identification of GPT-4o written peer reviews at low levels of false positive classifications. Our work reveals the difficulty of accurately identifying AI-generated text at the individual review level, highlighting the urgent need for new tools and methods to detect this type of unethical application of generative AI.

## 1   Introduction

Recent advancements in large language models (LLMs) have enabled their application to a broad range of domains, where LLMs have demonstrated the ability to produce plausible and authoritative responses to queries even in highly technical subject areas. These advancements have coincided with a surge in interest in AI research, resulting in large increases in paper submissions to leading AI conferences (Table S1). Consequently, workloads for peer reviewers of such conferences have also increased significantly, which could make LLMs an appealing tool for lessening the burden of fulfilling their peer review obligations.

Despite their impressive capabilities, the use of LLMs in the peer review process raises several ethical and methodological concerns which could compromise the integrity of the publication process. Reviewers are selected based on their expertise in a technical domain related to a submitted manuscript, which is necessary to critically evaluate the proposed research. Offloading this responsibility to an LLM circumvents the role that reviewer selection plays in ensuring proper vetting of a manuscript. Furthermore, LLMs are prone to hallucination and may not possess the ability to rigorously evaluate research publications. Therefore, the use of LLMs in an undisclosed manner in peer review poses a significant ethical concern that could undermine confidence in this important process.

Motivating the need for detection tools to address this problem is the apparent increase in AI-generated text among peer reviews submitted to recent AI conferences. Figure 1 shows the proportion of reviews submitted to ICLR between 2019 and 2024 which are flagged as containing AI text using methods deployed in this study. There is a consistent upward trend in recent years which provides indirect evidence of the increasing use of LLMs in peer review writing, echoing a recent study which

estimated that 6.5% to 16.9% of peer reviews submitted to recent AI conferences might have involved substantial use of LLMs for task beyond simple grammar checks [1].

In this work, we investigate the suitability of various AI text detection methods for identifying LLM generations in the peer review process. While limited prior work has analyzed the presence of AI-generated text in peer reviews at the corpus level [1] (see Appendix A for further discussion of related work), our study is the first to investigate the detectability LLM generations at the individual review level, which is necessary to address this problem in practice. Specifically, we produce AI-generated peer reviews for papers submitted to AI conferences prior to the introduction of ChatGPT using different LLMs and prompting methods. We then evaluate multiple open-source and proprietary AI text detection models on their ability to distinguish real peer reviews collected for these conferences from our AI-generated peer reviews for the same papers.

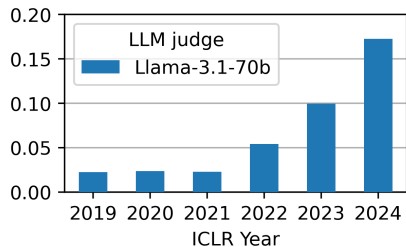

Figure 1: Proportion of reviews submitted to ICLR detected as AI-generated.

Our results show that all existing AI-text detection methods are limited in their ability to robustly detect AI-generated reviews while maintaining a low number of false positives. We propose an alternative approach to detecting AI-generated peer reviews by comparing the semantic similarity of a given review to a set of reference AI-generated reviews for the same paper, which surpasses the performance of all existing approaches in detecting GPT-4o written peer reviews. Our work demonstrate the challenging nature of detecting AI-written text in peer reviews and motivates the need for further research on methods to address this unethical use of LLMs in the peer review process.

## 2 Methodology

### 2.1 Data Collection

We used the OpenReview client API [2] to collect submitted manuscripts and their reviews for the ICLR conference from 2019 to 2024. The total numbers of submission and reviews are summarized in Table S1. We use two LLMs to generate AI peer reviews for these manuscripts, OpenAI's GPT-4o [3] and the open-source instruction-tuned Llama-3.1 (70b) [4], to generate AI peer reviews. Manuscripts are converted from PDF to Markdown format, excluding the Bibliography and Acknowledgement sections to focus on the core content relevant for review. Prompts used for generating AI reviews are adapted from Lu et al. [5] with two major changes. First, recognizing that LLM-generated text is sensitive to prompt variations, we introduce four distinct reviewer archetypes—"balanced," "nitpicky," "innovative," and "conservative"—to simulate the diversity of real-world peer review scenarios. Second, we provide the LLMs with the ICLR 2022 reviewer guidelines [6] with a minor modification. The ICLR guideline emphasize general instructions about how to approach peer review, rather than focusing on rubric and scoring scales as in the NeurIPS reviewer form. Our complete prompts are included in Appendix F. We collect a total of 16,000 AI-generated reviews for 500 random submissions for each ICLR conference of year from 2021-2024. Each AI-generated review contains five sections, and they were combined for detection tasks (see Appendix C for details).

### 2.2 AI-Generated Text Detection Methods

**Open-source AI text detection models.** We utilize two supervised fine-tuned pretrained language models. The first is from [7], where the author collect a Human ChatGPT Comparison Corpus (HC3) consisting of question-answer pairs created by human experts and generated by ChatGPT. The authors trained two Roberta-based models [8]: one detects whether answers are human or AI-generated using paired data, while the other evaluates single answers. We use the latter for our assessment. The second model is a Longformer [9]. As tested in [10], this model has demonstrated improved detection AUROC and generalization performance on the large-scale MAGE testbed, which includes eight distinct writing tasks, surpassing other methods such as GLTR [11], FastText [12], and DetectGPT [13]. For both models, we calculate the probability of each sentence in the review, averaging these to determine the final probability of the entire review being classified as AI-generated.

|  | FPR = 0.05 | | FPR = 0.20 | |
|---|---|---|---|---|
|  | GPT-4o Reviews | Llama Reviews | GPT-4o Reviews | Llama Reviews |
| GPT-4o Judge | - | - | 0.8040 | 0.9465 |
| Llama 3.1 Judge | 0.2390 | 0.7637 | - | - |
| Originality AI API | 0.5856 | **0.9985** | **0.9989** | **1.0000** |
| RoBERTa Classifier | 0.1855 | 0.8105 | 0.5110 | 0.9180 |
| Longformer Classifier | 0.4570 | 0.7860 | 0.8100 | 0.9375 |
| GPT-4o Anchor (ours) | **0.9670** | 0.8215 | 0.9985 | 0.9550 |
| Llama 3.1 Anchor (ours) | 0.9165 | 0.8880 | 0.9880 | 0.9675 |

Table 1: Comparison of true positive rates (TPR) for different AI detection methods applied to ICLR 2022 reviews. Since LLM judges only provide binary decisions instead of probability scores, they have fixed false positive rate (FPR) thresholds: 0.05 (GPT-4o judge) and 0.20 (Llama-3.1 judge). We use these two for the other methods and get the true positive rates (TPR).

**Originality AI API.** The Originality AI API is a commercial AI text detection service, and some studies have reported its high performance [14, 15]. It returns an AI score ranging between 0 and 1 which indicates the model's confidence that an input text was AI generated. While values greater than 0.5 indicate that the model has more confidence that an input text is AI-written than human-written, a higher threshold on the AI score may provide a more desirable balance between true positive and false positive classifications. Therefore, we investigate the trade-off between true positive and false positive classifications using a range of different thresholds on the returned score.

**LLM-based detection.** We employ the LLM-as-a-judge approach [16], utilizing both GPT-4o and Llama-3.1-70B models. The LLMs are instructed to provide two outputs: a binary decision ("human" or "AI") and a rationale for their decision, encouraging chain-of-thought reasoning.

**Anchor Embeddings Detection.** We propose a new method aiming to detect AI-generated reviews by comparing their semantic similarty to an AI-generated review for the same paper, which we refer to as the "Anchor Review". Specifically, we use GPT-4o or Llama3-70B to generate a review given a paper. We use simple prompt without any prior knowledge of prompts that used for generating the tested AI reviews. For each review we want to test, we use an off-the-shelf embedding model to create vector representations of both the Anchor Review and the test review. We then compute the cosine similarity between these two embeddings. The higher the similarity score, the more likely it is that the test review was generated by AI. By setting a threshold on this similarity score, we can classify reviews as either human-written or AI-generated at varying levels of sensitivity.

## 3   Results

**AI review detectability.** Table 1 compares the TPR of different detection methods on ICLR 2022 reviews when calibrated for a FPR of 0.05 and 0.20. As there is no threshold to tune the sensitivity of the GPT-4o and Llama 3.1 Judge models, we report only the single TPR and FPR for the binary classifications produced by these detectors. While GPT-4o is able to identify its own peer reviews over 80% of the time and those written by Llama 3.1 nearly 95% of the time, this comes at the cost of flagging 20% of all human reviews as AI-written. In contrast, the Originality AI API and our GPT-4o and Llama 3.1 Embedding models correctly identify nearly all GPT-4o written reviews at an identical FPR. At a lower (and more practiacal) FPR of 0.05, our GPT-4o and Llama 3.1 Embedding methods perform the best at detecting GPT-4o written reviews, identifying 97% and 95% of all such reviews (respectively). These methods also correctly identify 87-90% of Llama written peer reviews, while the Originality AI API identifies nearly all Llama 3.1 written reviews but is significantly worse at detecting GPT-4o written reviews. The RoBERTa and Longformer classifiers perform the worst among evaluated methods at both FPR levels. While we mainly present the results of 2022 year reviews, we provide additional results for ICLR reviews from 2021, 2023, and 2024 for other detection models in Appendix B.

Figure 2 provides area under the curve (AUC) plots for threshold-based classification methods which can be tuned for sensitivity, calculated separately for GPT-4o written peer reviews (Figure 2a) and Llama-3.1 written peer reviews (Figure 2b) of ICLR 2022 papers. These plots illustrates the relationship between the TPR and FPR for different AI text classification thresholds. Our GPT-4o

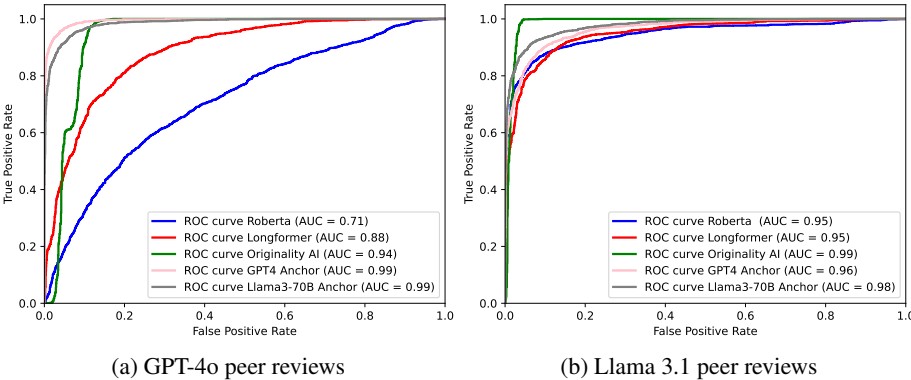

(a) GPT-4o peer reviews  (b) Llama 3.1 peer reviews

Figure 2: AUC plots for GPT-4o reviews (left) and Llama 3.1 reviews (right) of ICLR 2022 papers, calculated using five different AI text detection models.

and Llama-3.1 Embedding models have the highest AUC for GPT-4o written reviews, whereas the Originality AI API has the highest AUC for Llama 3.1 written reviews. Most Llama 3.1 reviews are correctly classified as AI-written by all methods at relatively low FPR values. In contrast, GPT-4o review detections are much lower at small FPR values for models other than our embedding-based approach; for example, the Originality AI API can detect only a little over half of GPT-4o written peer reviews when the classification threshold is calibrated to a FPR of 0.05. This shows how existing AI text detection methods struggle to consistently detect peer reviews generated by GPT-4o without also triggering a high number of false positives for human-written reviews.

**Analysis of LLM judge justifications.** To understand the decision basis of LLMs, we present findings from our analysis on the rationales behind LLM's decisions. We instructed an LLM to provide a brief justification for each decision. To identify the most representative samples, we first cluster the GPT-4o judge's rationale texts using t-SNE, using the SFR-Embedding model[17]. The t-SNE plot demonstrates that the rationales for AI and human decisions are well-separated (Figure S3), indicating systematic differences in them. Then, we pick the top 30 samples and the bottom 30 samples along the first dimension and identify common themes in the decision rationale (Table S4).

These findings suggest that the GPT-4o judge primarily relies on stylistic and content depth cues to distinguish between AI-generated versus human-written reviews, mainly through contrasting attributes. For example, the rationales for the AI label are characterized by repetitiveness, lack of coherence, formal tone, generic critique, and lack of specificity—issues commonly observed in LLM-generated texts. In contrast, human-written reviews exhibit the opposite attributes, such as nuanced discussions, subjective opinions, conversational tone, specific critique, and detailed references.

Some of these findings raise concerns about the use of LLM-as-a-judge style detection models, or any model that takes advantage of similar features. Echoing the findings of Liang et al. [18], "non-native language use" emerged as one of the reasons for AI labels, raising fairness concerns of discriminating non-native English speakers. More broadly, even native speakers with unconventional writing styles (e.g., "awkward sentence structure") may be unfairly penalized. These issues should be considered in future AI detector designs to minimize unintended biases.

## 4 Conclusion

In this work, we showed that existing approaches for detecting AI text are poorly suited to the challenging problem of identifying AI-generated peer reviews. While high detection rates are possible with existing methods, this comes at the cost of relatively high rates of falsely identifying human-written reviews as containing AI text, which must be minimized in practice. We proposed a new approach which intentionally generates AI-written reviews for a given paper to serve as a basis for comparing semantic similarity to other evaluated reviews, achieving much higher accuracy in identifying GPT-4o written peer reviews while maintaining a low level of false positives. Our results demonstrate the promise of this approach while also motivating the need for further research on methods for detecting unethical applications of LLMs in the peer review process.

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

| ICLR Year | 2019 | 2020 | 2021 | 2022 | 2023 | 2024 |
|---|---|---|---|---|---|---|
| Submissions | 1,565 | 2,561 | 2,452 | 2,587 | 3,793 | 7,306 |
| Reviews | 4,764 | 7,775 | 9,479 | 10,080 | 14,355 | 28,028 |

Table S1: Number of submissions and peer reviews for the ICLR conference from 2019 to 2024, based on data scraped from `OpenReview.net` in September 2024.

## A   Related Work

**AI generated text detection.**    Early approaches to AI-generated text detection predominantly framed the task as a binary classification problem, where the objective was to determine whether a given text was human-written or machine-generated [13, 19–21]. For instance, Solaiman et al. [22] employed a bag-of-words model combined with logistic regression to distinguish GPT-2 generated web articles from human-authored content. Several studies focused on fine-tuning pre-trained language models such as RoBERTa [23] to improve detection accuracy [11, 24, 25]. These detectors leveraged the internal representations of language models to differentiate human and machine text. Concurrently, researchers explored zero-shot detection techniques that avoided the need for additional training, with methods relying on features like perplexity or entropy to detect machine-generated text [11, 26]. Moreover, work by Zellers et al. [24] introduced neural networks specifically trained to spot AI-generated misinformation, while Ippolito et al. [26] further advanced zero-shot detection using likelihood-based methods. Another line of research highlighted the use of linguistic patterns, syntactic structures, and word distributions to identify AI-generated content without direct fine-tuning, thus improving adaptability to new models [11, 25]. More recent studies explored watermarking and cryptographic signals embedded in LLM-generated text to enhance detection, such as the method proposed by Mitchell et al. [13], which enables proactive identification.

**AI-assisted peer review.**    There has been substantial research examining the implications of using generative AI in conference peer review processes [1, 27–32]. Liang et al. [1] introduced a maximum likelihood method to assess the influence of large language models (LLMs) on reviewing practices at the corpus level. They also conducted a large-scale empirical study [27] to evaluate the effectiveness of LLM-generated feedback compared to human reviews, revealing insights into its utility and limitations in peer review. In another effort, Tan et al. [28] developed a comprehensive dataset that simulates the peer review process as a multi-turn dialogue, incorporating the roles of reviewers, authors, and meta-reviewers to explore the dynamics of review interactions. Zhou et al. [29] focused on evaluating the reviewing capabilities of LLMs such as GPT-3.5 and GPT-4o through 196 multiple-choice questions related to reviewing tasks, providing insights into their strengths and weaknesses in academic review settings. To enhance the quality of reviews, Tyser et al. [30] proposed the OpenReviewer tool, which allows authors or reviewers to submit papers and receive automatic reviews, enabling iterative improvements before actual submission or peer review. Kuznetsov et al. [31] explored the core challenges in the peer review process and argued for a more integrated, transparent use of LLMs in reviewing to overcome these limitations. Additionally, Mosca et al. [32] introduced a benchmark designed to distinguish between human-written and machine-generated scientific papers, helping to address concerns about the integrity of AI-generated content in academic settings. In contrast, our work focuses on the utility of AI text detection methods in identifying LLM generated content in the peer review process.

| Year | GPT-4o | | | Llama-3.1-70b | | |
|---|---|---|---|---|---|---|
| | TPR (GPT) | TPR (Llama) | FPR (human) | TPR (GPT) | TPR (Llama) | FPR (human) |
| 2021 | 0.8070 | 0.9405 | 0.1483 | 0.2407 | 0.7769 | 0.0229 |
| 2022 | 0.8040 | 0.9465 | 0.1957 | 0.2390 | 0.7637 | 0.0540 |
| 2023 | 0.7923 | 0.9446 | 0.2973 | 0.2005 | 0.7814 | 0.0995 |
| 2024 | 0.8440 | 0.9530 | 0.4677 | 0.2447 | 0.7711 | 0.1727 |

Table S2: Comparison of true positive rates (TPR) and false positive rates (FPR) for two different LLM judges across various ICLR conference years.

## B   Additional detection results for other ICLR conference years

The LLM judge-based results for ICLR 2021–2024 are shown in Table S2. The true positive rates (TPR) for AI-generated reviews are consistent overall from 2021 to 2024, regardless of which LLM was used to generated the reviews or to evaluate them. In contrast, a clear trend is observed for human-written reviews: the false positive rate (FPR), which represents the proportion of human-written reviews incorrectly classified as AI-generated, increases almost exponentially for both GPT-4o and Llama-3.1 judges. Given the stability of TPR for AI-generated reviews over the same period and considering that 2021-2022 marks the time when LLMs started to gain popularity, the rise in FPR is probably driven by the growing use of LLMs in recent years, leading to more peer reviews that are fully or partially LLM-generated.

## C   Impact of text formatting.

In our dataset, AI-generated reviews consistently included the five sections: "Summary," "Strengths," "Weaknesses," "Questions," and "Limitations." In contrast, human reviews varied in their section structures due to differing review templates across conference years (e.g., the 2021, 2022, 2023, and 2024 templates have one, three, four and four sections, respectively). While we found large variations in detection performance across different sections (Figure S1), we used the aggregated text (i.e., combining all sections) in our study. Moreover, we found that the detection results are sensitive to the formatting of the combined reviews, e.g., section headings and itemized lists (Figure S2). Consequently, we chose not to standardize review section formatting across different conferences to simplify the methods and to avoid introducing additional sources of bias by converting irregular, free-formed human reviews into a standardized structure.

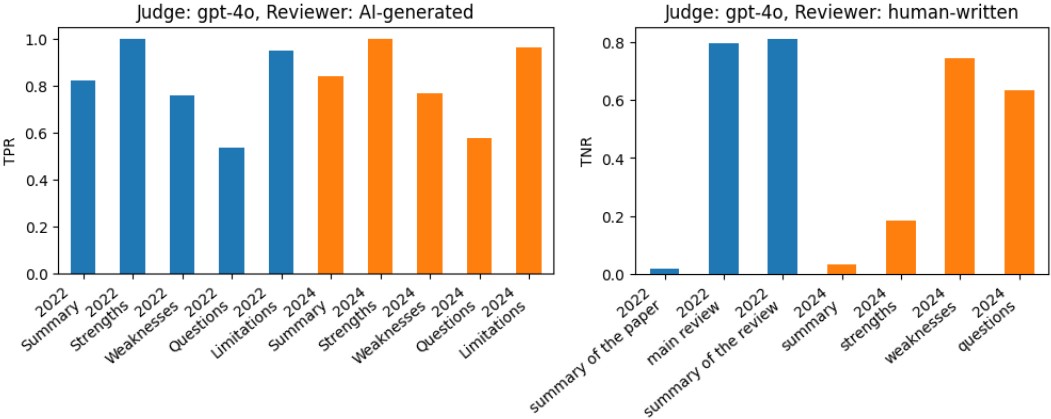

Figure S1: (left) true positive rate (TPR) of AI-generated reviews and (right) true false rate (TFR) of human-written reviews, based on our GPT-4o judge. The evaluation is done for each section, as indicated in the x-axis labels. Blue bars are for ICLR2022, and orange bars are for ICLR2024.

## D   Accuracy of estimating AI classification threshold based on desired FPR

In practice, an appropriate threshold for the AI text classifiers could be set based on a desirable maximum false positive rate (FPR) for human reviews. It may be necessary to allow only a relatively small false positive rate (e.g., $< 5\%$) if such reviews are to be manually investigated. In Table S3, we estimate the actual FPR for different targeted FPR values for the Originality AI API using 5-fold cross validation. These results show that estimating FPR using a withheld set of human reviews is relatively accurate. Assuming a 5% FPR is acceptable, a threshold of 0.51 for the AI score could be used. While this threshold would correctly detect 99.9% of peer reviews written by Llama 3.1, it would only detect 58.6% of those written by GPT-4. Decreasing the FPR to 1% with a score threshold of 0.97 would detect only 0.1% of GPT-4o generated peer reviews. This highlights the challenge of reliably detecting AI-written content in peer review without falsely classifying human-authored reviews and the inadequacy of current commercial solutions for this problem.

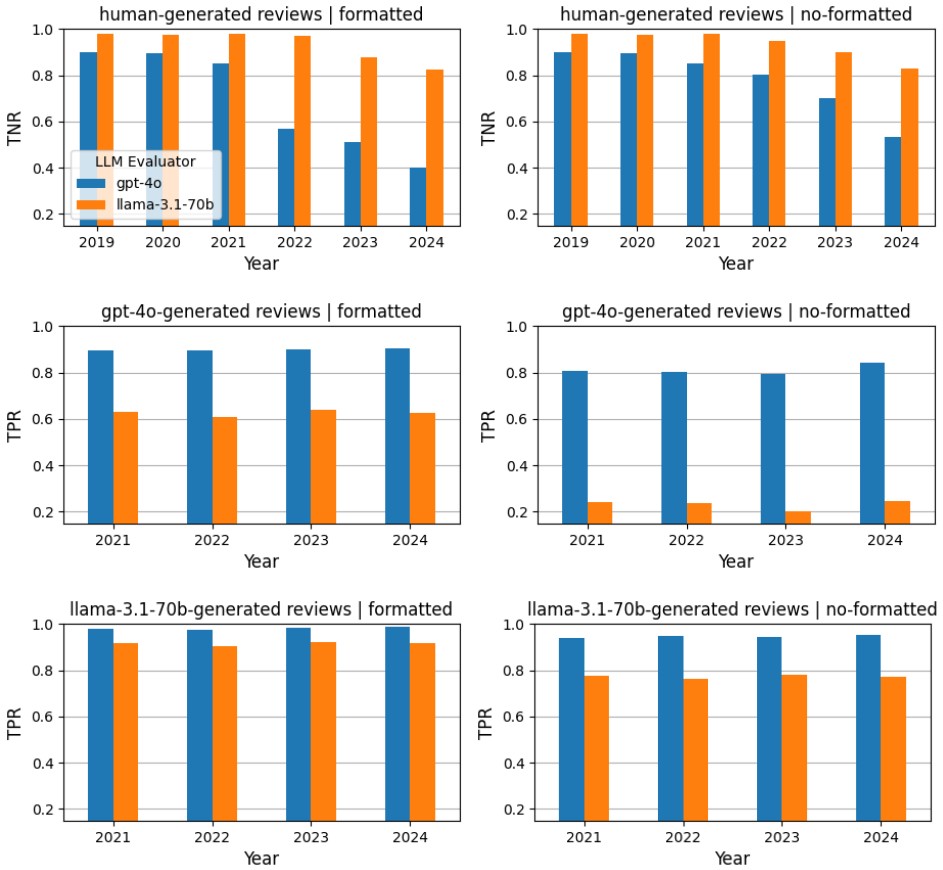

Figure S2: Comparison of LLM-as-a-judge performance with and without text formatting. Each row represents a different source of review texts: (top) human-written, (middle) GPT-4o-generated, and (Bottom) Llama-3.1-70b-generated. Each column compares the effect of formatting: (left) formatted reviews (with section headings and itmemization) and (right) non-formatted reviews. GPT-4o (blue) Llama-3.1-70b (orange) are used as an LLM judge. Note that AI-generated reviees are labeled as positive. For example, True negative rate (TNR) represents the proportion of human-written reviews correctly identified as human-written (that is, not AI-generated), while true positive rate (TPR) represents the proportion of AI-generated reviews correctly classified. Collectively, TNR and TPR are the proportions of reviews that were correctly classified according to their original labels.

| Target FPR | Actual FPR | Score Threshold |
|---|---|---|
| 0.01 | $0.0099 \pm 0.0096$ | $0.9658 \pm 0.0084$ |
| 0.05 | $0.0496 \pm 0.0186$ | $0.5159 \pm 0.0143$ |

Table S3: Targeted false positive rate (FPR) vs. actual for Originality AI API score thresholds estimated using 5-fold cross validation on human-written reviews. Values for Actual FPR and Score Treshold are the mean $\pm$ standard deviation.

# E   Representative Examples of LLM Reasoning

A few samples of actual reasoning provided by the GPT-4o judge are listed here. These samples were chosen based on the t-SNE plot of the ICLR2024 human-written reviews (Figure S3), with the "AI" decision sample representing the lowest value on the first dimension, and the "human" decision sample representing the highest value.

Reasoning for "AI" decisions:

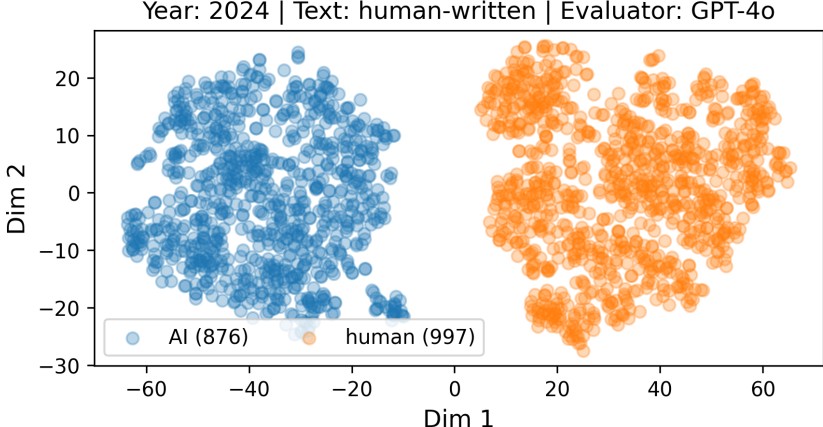

Figure S3: t-SNE visualization of embeddings from the GPT-4o judge on the ICLR 2024 human-written reviews. Each point represents an embedding of the decision rationale output of the LLM, which contains a few sentences (see the example in Section E). Blue indicates reviews classified as AI-generated, and orange indicates reviews classified as human-written. The t-SNE was performed with a target dimension of 2 and a perplexity value of 30.

| LLM Decision | Common reasons for decision |
| --- | --- |
| AI | Repetitive phrasing, lack of coherence, awkward sentence structure, grammatical/spelling errors, typos, formal tone, inconsistent evaluations, inconsistent formatting, generic critique, lacking specificity, non-native language use, abrupt topic shifts, vague reference to literature, technical jargons without explanation or context |
| human | Subjective opinions, critical analysis, specific reference to figures or academic literature, detailed critique, rhetorical questions, conversational tones, personal insights, detailed technical analysis, nuanced discussion, expression of uncertainty, specific questions, deep understanding of subject matters, comparison to existing literature, personal reflections, informal/conversational language, first-person pronouns |

Table S4: Summary of common reasons given by the LLM judge for classifying peer reviews as either AI-generated or human-written. The reasons include language features, content depth and coherence, and specific stylistic elements that indicate whether the text was generated by an AI or written by a human.

- The text exhibits characteristics typical of AI-generated content, such as repetitive phrasing, awkward sentence structures, and a lack of cohesive flow. The critique points are structured in a mechanical manner, and there are grammatical errors that suggest automated generation. The text lacks the nuanced language and varied sentence structures often found in human writing, especially in academic reviews or critiques.

- The text exhibits characteristics typical of AI-generated content, such as repetitive phrases, lack of coherence, and inconsistent focus. The critique of the paper is vague and lacks depth, which is common in AI-generated text. Additionally, the request to 'check the weaknesses' at the end is abrupt and uncontextualized, suggesting automated generation rather than human authorship.

- The text exhibits characteristics typical of AI-generated content, such as repetitive structure, lack of cohesive flow, and awkward phrasing (e.g., 'Write is good and easy to follow'). The analysis is presented in a list format with inconsistent numbering, and

there are grammatical errors like 'sometime does not make sense' and 'these example are.' These features suggest automated generation rather than human authorship.

- The text contains several indicators of AI generation: awkward phrasing ('Richful experiments'), inconsistent grammar ('althrough'), and a lack of coherent structure. The technical content is presented in a way that lacks clarity and precision, which is common in AI-generated text attempting to mimic academic writing. Additionally, the text includes a mix of technical jargon and incomplete explanations, which suggests an AI-generated attempt to simulate expert knowledge.

- The text exhibits characteristics typical of AI-generated content, such as repetitive phrasing, awkward sentence structures, and a lack of natural flow. The use of phrases like 'the proposed model need to train a model' and 'hasn't been not fully evaluated' suggests a lack of proofreading and human-like coherence. Additionally, the text lacks specific details and depth that a human expert might include, especially in a technical paper review.

Reasoning for "human" decisions:

- The text includes subjective opinions, critical analysis, and specific concerns about the dataset and experimental results, which are indicative of human authorship. The use of phrases like 'can not convince me' and detailed critique of methodologies suggest a personal perspective and depth of analysis typical of human writing rather than AI-generated content.

- The text contains critical analysis and specific references to academic literature, which are indicative of human authorship. The writing style includes subjective opinions, detailed critique, and nuanced understanding of complex topics like external validity and collider bias, which are less typical of AI-generated text. The text also includes rhetorical questions and a conversational tone, further suggesting a human author.

- The text contains subjective analysis, personal insights, and critiques typical of human writing. It discusses specific models and methods in a nuanced manner, referencing personal research and experimentation. The use of phrases like 'From my own research' and 'It wasn't surprising to find' indicates personal experience and opinion, which are characteristic of human authorship rather than AI-generated content.

- The text includes detailed technical analysis, specific critiques, and a request for clarification, which are indicative of human authorship. The nuanced discussion of biases, causal graphs, and experimental concerns suggests a deep understanding of the subject matter, typical of a human researcher. The text also contains subjective expressions like 'It seems to me' and 'I believe,' which are less common in AI-generated content.

- The text exhibits characteristics typical of human writing, such as critical analysis, subjective opinions, and specific references to academic works. It discusses weaknesses and relevancy concerns in a nuanced manner, which suggests a deep understanding of the subject matter. The text also includes specific citations and comparisons to existing literature, indicating a level of expertise and context that is more aligned with human authorship rather than AI-generated content.

# F Prompts

## F.1 Prompt for LLM judges

System prompt:

```
You will be given a text. Your task is to determine if the text is either
written by a human or generated by a large language model (LLM).

First, read and analyze the given text rigorously.

Then, if you think any portion of the text is written by AI (that is, large
language models), type "AI". Type "human" only if you think the entire text
is written by a human.

Write your answer in the following JSON format, and output only the JSON
code block without any additional text or explanations outside of it:

```json
{
  "Result": "<'human' or 'AI'>",
  "Rationale": "<Provide a clear and concise justification for your
  decision. Ensure your explanation highlights specific features that
  influenced your decision. Do not use more than 100 words.>"
}
```
```

User prompt:

```
Here is the text you are asked to assess:

```
{review}
```
```

## F.2 Prompt for generating reviews

Prompts used for generating AI reviews were adapted from Lu et al. [5] with two major changes. First, we introduced four archetypes of peer reviewers to generate a diverse set of reviews, capturing the varying characteristics across different reviews. Second, instead of the NeurIPS reviewer guideline, we provide the LLMs with the ICLR 2022 reviewer guideline[1] with a minor modification. The ICLR guideline emphasize general instructions about how to approach the peer review, rather than focusing on rubric and scoring scales as in the NeurIPS reviewer form.

Reviewer type prompt:

- <balanced>
  ```
  You provide fair, balanced, thorough, and constructive feedback,
  objectively highlighting both the strengths and weaknesses of
  the paper.  You maintain a high standard for research in your
  decision-making process.  However, even if your decision is to
  reject, you offer helpful suggestions for improvement.
  ```
- <nitpicky>
  ```
  You are a perfectionist who meticulously examines every aspect
  of the paper, including minor methodological details, technical
  nuances, and formatting inconsistencies.  Even if a paper presents
  novel ideas or significant contributions, you may still recommend
  rejection if you identify a substantial number of minor flaws.  Your
  ```

---

[1]https://iclr.cc/Conferences/2022/ReviewerGuide

stringent attention to detail can sometimes overshadow the broader
significance of the work in your decision-making process.

- • <innovative>
  You highly value novelty and bold approaches, often prioritizing
  novel ideas over methodological perfection. While you maintain high
  standards, you are willing to overlook minor flaws or incomplete
  validations and may accept the paper, if the paper introduces a
  significant new concept or direction. Conversely, you tend to be
  less enthusiastic about papers that, despite thorough methodology
  and analysis, offer only incremental improvements, and may recommend
  rejection for such submissions.
- • <conservative>
  You generally prefer established methods and are skeptical of
  unproven (that is, new or unconventional) approaches. While you
  maintain high standards and rigor, you are critical of papers
  presenting new ideas without extensive evidence and thorough
  validation against established baselines. You place significant
  emphasis on methodological soundness and are cautious about
  endorsing innovations that haven't been rigorously tested.

System prompt:

You are an AI researcher tasked with reviewing a paper submitted to a
prestigious AI research conference.
{reviewer_type}
You will be provided with the paper to evaluate.
Follow the provided review guidelines and submit your assessment using the
specified response template.

## Review guideline
{iclr_2022_guideline}

## Response template
Respond in the following format:

THOUGHT:
<THOUGHT>

REVIEW JSON:
```json
<JSON>
```

In <THOUGHT>, you provide the summary of your review.
First, briefly discuss your intuitions and reasoning for the evaluation.
Detail your high-level arguments, necessary choices, and desired outcomes
of the review.
Do not make generic comments here, but be specific to your current paper.
Treat this as the note-taking phase of your review.

In <JSON>, provide the review in JSON format with the following fields in
the order:
- "Summary": One paragraph including three parts --- a detailed summary of
the paper content and its contributions, detailed justifcation for your
decision, and decision recommendation.
- "Strengths": A list of strengths of the paper. Each item should be a ful
sentence.
- "Weaknesses": A list of weaknesses of the paper. Each item should be a
ful sentence.

- "Questions": A set of clarifying questions to be answered by the paper authors. Each item should be a ful sentence.
- "Limitations": A set of limitations and potential negative societal impacts of the work. Each item should be a ful sentence.
- "Ethical Concerns": A Boolean value indicating whether there are ethical concerns.
- "Originality": A rating from 1 to 4 (low, medium, high, very high).
- "Quality": A rating from 1 to 4 (low, medium, high, very high).
- "Clarity": A rating from 1 to 4 (low, medium, high, very high).
- "Significance": A rating from 1 to 4 (low, medium, high, very high).
- "Soundness": A rating from 1 to 4 (poor, fair, good, excellent).
- "Presentation": A rating from 1 to 4 (poor, fair, good, excellent).
- "Contribution": A rating from 1 to 4 (poor, fair, good, excellent).
- "Overall": A rating from 1 to 10 (very strong reject to award quality).
- "Confidence": A rating from 1 to 5 (low, medium, high, very high, absolute).
- "Decision": A decision that must be one of the following: Accept, Reject.

For the "Decision" field, don't use Weak Accept, Borderline Accept, Borderline Reject, or Strong Reject. Instead, only use Accept or Reject. This JSON will be automatically parsed, so ensure the format is precise.

User prompt:

Here is the paper you are asked to review:
```
{text}
```

ICLR 2022 Reviewer guideline:

## Step-by-step Review Instructions
A review aims to determine whether a submission will bring sufficient value to the community and contribute new knowledge. The process can be broken down into the following main reviewer tasks:

1. Read the paper: It is important to carefully read through the entire paper, and to look up any related work and citations that will help you comprehensively evaluate it. Be sure to give yourself sufficient time for this step.

2. While reading, consider the following:
  - Objective of the work: What is the goal of the paper? Is it to better address a known application or problem, draw attention to a new application or problem, or to introduce and/or explain a new theoretical finding? A combination of these? Different objectives will require different considerations as to potential value and impact.
  - Strong points: is the submission clearly written, technically correct, experimentally rigorous, reproducible, does it present novel findings (e.g. theoretically, algorithmically, etc.)?
  - Weak points: is it weak in any of the aspects listed above?
  - Be mindful of potential biases and try to be open-minded about the value and interest a paper can hold for the entire research community, even if it may not be very interesting for you.
  - Originality: Are the tasks or methods new? Is the work a novel combination of well-known techniques? (This can be valuable!) Is it clear how this work differs from previous contributions? Is related work adequately cited?

- Quality: Is the submission technically sound? Are claims well supported (e.g., by theoretical analysis or experimental results)? Are the methods used appropriate? Is this a complete piece of work or work in progress? Are the authors careful and honest about evaluating both the strengths and weaknesses of their work?
- Clarity: Is the submission clearly written? Is it well organized? (If not, please make constructive suggestions for improving its clarity.) Does it adequately inform the reader? (Note that a superbly written paper provides enough information for an expert reader to reproduce its results.)
- Significance: Are the results important? Are others (researchers or practitioners) likely to use the ideas or build on them? Does the submission address a difficult task in a better way than previous work? Does it advance the state of the art in a demonstrable way? Does it provide unique data, unique conclusions about existing data, or a unique theoretical or experimental approach?
- Ethical concerns: If there are ethical issues with this paper, please flag the paper for an ethics review.

3. Answer three key questions for yourself, to make a recommendation to Accept or Reject:
- What is the specific question and/or problem tackled by the paper?
- Is the approach well motivated, including being well-placed in the literature?
- Does the paper support the claims? This includes determining if results, whether theoretical or empirical, are correct and if they are scientifically rigorous.

4. Write your initial review, organizing it as follows:
- Summarize what the paper claims to contribute. Be positive and generous.
- List strong and weak points of the paper. Be as comprehensive as possible.
- Clearly state your recommendation (accept or reject) with one or two key reasons for this choice.
- Provide supporting arguments for your recommendation.
- Ask questions you would like answered by the authors to help you clarify your understanding of the paper and provide the additional evidence you need to be confident in your assessment.
- Provide additional feedback with the aim to improve the paper. Make it clear that these points are here to help, and not necessarily part of your decision assessment.

5. Provide a numeric score for the following aspects of the paper:
- "Overall": Provide an overall score for this submission. Choices:
  10, Absolute accept. Top 5% of accepted papers, seminal paper
  9, Strong accept. Top 15% of accepted papers
  8, Clear accept. Top 50% of accepted papers
  7, Accept. Good paper
  6, Weak accept. Marginally above acceptance threshold
  5, Weak rejection. Marginally below acceptance threshold
  4, Rejection. Ok but not good enough
  3, Clear rejection
  2, Strong rejection
  1, Absolute rejection. Trivial or wrong
- "Confidence": Provide a confidence score for your assessment of this submission. Choices:

5, You are absolutely certain about your assessment. You are very
familiar with the related work and checked the math/other details
carefully.
4, You are confident in your assessment, but not absolutely certain. It
is unlikely, but not impossible, that you did not understand some parts
of the submission or that you are unfamiliar with some pieces of
related work.
3, You are fairly confident in your assessment. It is possible that you
did not understand some parts of the submission or that you are
unfamiliar with some pieces of related work. Math/other details were
not carefully checked.
2, You are willing to defend your assessment, but it is quite likely
that you did not understand central parts of the submission or that you
are unfamiliar with some pieces of related work. Math/other details
were not carefully checked.
1, Your assessment is an educated guess. The submission is not in your
area, or the submission was difficult to understand. Math/other details
were not carefully checked.
- "Soundness": Assign the paper a numerical rating on the following scale
to indicate the soundness of the technical claims, experimental and
research methodology and on whether the central claims of the paper are
adequately supported with evidence. Choices:
  4, excellent
  3, good
  2, fair
  1, poor
- "Presentation": Assign the paper a numerical rating on the following
scale to indicate the quality of the presentation. This should take into
account the writing style and clarity, as well as contextualization
relative to prior work. Choices:
  4, excellent
  3, good
  2, fair
  1, poor
- "Contribution": Assign the paper a numerical rating on the following
scale to indicate the quality of the overall contribution this paper
makes to the research area being studied. Are the questions being asked
important? Does the paper bring a significant originality of ideas and/or
execution? Are the results valuable to share with the broader research
community. Choices:
  4, excellent
  3, good
  2, fair
  1, poor

6. General points to consider:
  - Be polite in your review. Ask yourself whether you'd be happy to
  receive a review like the one you wrote.
  - Be precise and concrete. For example, include references to back up any
  claims, especially claims about novelty and prior work.
  - Provide constructive feedback.
  - It's also fine to explicitly state where you are uncertain and what you
  don't quite understand. The authors may be able to resolve this in their
  response.
  - Don't reject a paper just because you don't find it ''interesting''. This
  should not be a criterion at all for accepting/rejecting a paper. The
  research community is so big that somebody will find some value in the
  paper (maybe even a few years down the road), even if you don't see it
  right now.

