# OpenReview forum: "Is Your Paper Being Reviewed by an LLM? Investigating AI Text Detectability in Peer Review"
_NeurIPS.cc/2024/Workshop/SafeGenAi — SafeGenAi Poster_

### Official Review · Reviewer_3x8H · 2024-10-08
**Review of AI-text detection in Peer Reviews; Strong Introduction/Conclusion but needs more clarity for the methodology**

**Rating:** 5
**Confidence:** 4

**Review:**

This paper discusses concerns about LLM usage in peer reviews. The authors also claim that current detection methods do not work well and suggest an alternative approach based on semantics. This is an interesting topic to consider - AI text detection typically focuses on student essays in the academic domain but not on peer reviews. This is a crucial topic to consider as LLM usage proliferates across journals and conferences.

## Pros
- The authors propose a strong case for researching LLM usage in peer reviews, as seen in their introduction.
- The authors analyze the limitations of using LLMs as a judge - they mention certain drawbacks such as potential biases against non-native English speakers.


## Questions regarding the paper
- The code/data generated is unavailable (or at least not found easily). It would be helpful to future readers to gain access to the code and the dataset.
- Train/test splits were not defined. How large was the training dataset? What about the test set? Were the test sets balanced.
- Were the Longformer and RoBERTa classifiers trained on other AI-text detection datasets? Were they trained on your AI review dataset? Do the previously mentioned datasets, HC3 and MAGE, cover AI reviews?
- An example for the anchor embeddings would make the section easier to understand. Additionally, please explain which data was used for vector comparison. Was it data from your generated dataset, or did you prompt the respective LLM for another review (separate from the 16,000 reviews)?
- Are there any limitations on your dataset? For example, does your data only cover reviews from computer science-related submissions?
- The authors mention four archetypes - does this affect detection success?

The introduction and conclusion are relatively strong, but the methodology section needs additional information to clarify some doubts. A clearer methodology section can also help contextualize the results better for others.

---

### Official Review · Reviewer_oxtH · 2024-10-09
**The work  discusses the ability of AI tools to correctly identify whether or not the peer review submitted by a reviewer is AI generated. The work demonstrates effectivenesss of a simple approach based on comapring semantic similarty of  AI-generated review with the test review.**

**Rating:** 6
**Confidence:** 5

**Review:**

**Summary**

The work presents a new method to detect AI generated reviews by comparing the semantic similarity of the review to be evaluated with the AI generated review for the same paper, i.e., an AI review is deliberately generated for the paper and is compared against the given review using cosine similarity. This simple yet effective approach outperforms state of the art classifiers specially with GPT-4o generated reviews where proposed approach has >95% TPR while existing approaches have 18-60% TPR.

It would be useful to extend this work to include analogies with clustering algorithms. Can authors demonstrate a plot similar to the t-SNE plot presented for  GPT-4o judge in appendix for the proposed approach? This might help uncover underlying structures and relationships within the data that is leading to an effective performance of the proposed approach. It would also be interesting to see if a tree based model can be learned if the key features leading to high accuracy of the proposed approach are recovered?

While the proposed simple approach outperforms state-of-the-art methods, the paper lacks depth and analysis on interpreting the results.  Hence, I recommend weak accept.

**Strengths**
1. Paper is well-written and easy to follow. Paper proposes a simple approach which outperforms state-of-the-art models in identifying AI generated reviews on ICLR dataset.

**Weaknesses**
1. The paper lacks novelty. The proposed approach is straightforward where a review is generated using an LLM and is then used as a basis to evaluate a given review. More experiments should be done to identify the shortcomings of the proposed approach. How does the model perform on a different dataset? Is there any structure in GPT-4o dataset that the proposed approach is exploiting?
2. The paper uses cosine similarity but is there a different metric that was/ can be used?

---

### Official Review · Reviewer_MZvJ · 2024-10-09
**How do you define the AI peer reviews?**

**Rating:** 6
**Confidence:** 5

**Review:**

1. How do you conceptualize AI's role in the peer review process? It is understood that many academics meticulously read papers multiple times and jot down key observations. Following this, tools like ChatGPT are employed to refine the language based on these notes. For instance, a reviewer might draft a concise summary with five key sentences and ask ChatGPT to construct a comprehensive peer review. Could you clarify the extent of AI involvement? Is the final review output generated by AI, or does it merely assist in structuring the reviewer’s original thoughts? (For example, I utilize ChatGPT to structure this review. I don’t allow the AI to read the paper; instead, I input the main ideas from my comments, and the AI assists in organizing them into a coherent review.:)
2. Could you discuss the potential biases introduced by LLM-based detection models, particularly those that might disadvantage non-native English speakers or authors with unconventional writing styles? What strategies might be employed in future research to mitigate such biases?
3. The study concentrates on identifying LLM-generated text within individual reviews. However, it would be valuable to explore how these detection methods fare when applied across a broader corpus.